# Using Variational Quantum Algorithm to Solve the LWE Problem

**DOI:** 10.3390/e24101428

**Published:** 2022-10-08

**Authors:** Lihui Lv, Bao Yan, Hong Wang, Zhi Ma, Yangyang Fei, Xiangdong Meng, Qianheng Duan

**Affiliations:** 1State Key Laboratory of Mathematical Engineering and Advanced Computing, Zhengzhou 450001, China; 2Henan Key Laboratory of Network Cryptography Technology, Zhengzhou 450001, China

**Keywords:** quantum, LWE, QAOA, VQE, KYBER

## Abstract

The variational quantum algorithm (VQA) is a hybrid classical–quantum algorithm. It can actually run in an intermediate-scale quantum device where the number of available qubits is too limited to perform quantum error correction, so it is one of the most promising quantum algorithms in the noisy intermediate-scale quantum era. In this paper, two ideas for solving the learning with errors problem (LWE) using VQA are proposed. First, after reducing the LWE problem into the bounded distance decoding problem, the quantum approximation optimization algorithm (QAOA) is introduced to improve classical methods. Second, after the LWE problem is reduced into the unique shortest vector problem, the variational quantum eigensolver (VQE) is used to solve it, and the number of qubits required is calculated in detail. Small-scale experiments are carried out for the two LWE variational quantum algorithms, and the experiments show that VQA improves the quality of the classical solutions.

## 1. Introduction

Lattice theory is a classic subject in mathematical research, and it has critical applications in many fields such as the optimization problem and information coding. In 1996, Ajtai [1] proved that the worst-case hardness of the shortest vector problem (SVP) can be reduced to the hardness of SVP in a class of random lattices, thus providing provable security of lattice-based cryptosystems. Since then, various lattice-based cryptosystems are proposed, such as Ajtai-Dwork [2] and the Number Theory Research Unit [3].

In 2005, Regev proposed an encryption algorithm based on LWE [4]. Compared with previous lattice-based cryptosystems, the ciphertext size and key size of LWE-based cryptosystems are greatly reduced. Therefore, LWE began to be applied to many cryptographic primitives, such as Key-Dependent Message [5], Fully Homomorphic Encryption [6] and so forth. In July 2022, The National Institute of Standards and Technology completed the third round of the Post-Quantum Cryptography standardization process, and four candidate algorithms have been announced. Among them, the public-key encryption algorithm CRYSTALS-KYBER [7] and the digital signature algorithm CRYSTALS-Dilithium [8] are constructed based on the module-LWE problem. Therefore, analyzing LWE algorithms is important to the security of post-quantum cryptography.

The analysis methods of LWE can be classified into combinatorial methods, algebraic methods, lattice methods and the exhaustive search. The combinatorial method mainly refers to an extended application of the Gaussian elimination [9], but it requires a large number of samples. The algebraic method refers to the Arora-Ge algorithm [10], and the complexity is also exponential in the number of LWE dimensions. There are three main lattice methods: the dual method is used to attack decision-LWE instances by solving the short integer solution problem on the dual lattice [1]; the decoding method is used to directly solve the bounded distance decoding problem (BDD) on the original lattice [11,12]; the primary method is used to further reduce the BDD problem to the Unique-SVP problem [13,14,15]. The exhaustive search is not suitable for practical applications because of its high time complexity.

At the same time, VQA, such as QAOA [16], VQE [17], and FQE [18], has become the most suitable technology to achieve quantum advantage using noisy intermediate-scale quantum (NISQ) devices. Some works have studied how to solve hard lattice problems by VQA. Paper [19] analyzed the energy gaps between the first three excited states of the Hamiltonian when solving SVP with low dimension by quantum adiabatic computation. The conclusion in [19] inspired the use of QAOA to find the ground state. Ref. [20] calculated the number of qubits for special lattices and concluded that 1.5nlogn+n+log(det(L)) qubits sufficed to obtain the shortest vector of n-dimensional lattice L. Ref. [21] proposed to solve SVP by VQE and also pointed out that their algorithm was not limited to special lattices.

The work in this paper consists of two aspects. Firstly, we use QAOA to optimize the Nearest Plane algorithm and solve LWE. Secondly, inspired by Ref. [21], we propose a hybrid algorithm using VQE to attack LWE and calculate the number of qubits required to attack specific LWE cryptosystems. For the two LWE algorithm ideas, we conduct small-scale experimental simulations. The experiments show that QAOA improves the quality of classical solutions, and the quality of solutions obtained by VQE is at least equal to that of classical solutions when the memory is big enough.

## 2. Preliminary

### 2.1. Lattice Theory

Let $b_{1},b_{2},\ldots ,b_{n}\in R^{m}$ be a set of linearly independent vectors, and the lattice generated by $b_{1},b_{2},\ldots ,b_{n}$ is
$$\Lambda =L\left(b_{1},b_{2},\ldots ,b_{n}\right)=\left\{\alpha_{1}b_{1}+\alpha_{2}b_{2}+\ldots +\alpha_{n}b_{n}|\alpha_{1},\alpha_{2},\ldots ,\alpha_{n}\in Z\right\}.$$
In cryptography applications, the lattice dimension is *n*. Given a matrix $A\in Z_{q}^{m*n}$, the q-ary lattice refers to
$$\Lambda_{q}\left(A^{T}\right)=\left\{x\in Z^{m}|\exists y\in Z^{n},s.t.x\equiv yA^{T}modq\right\}.$$

For a lattice L and its basis matrix $B=[b_{1},b_{2},\ldots ,b_{n}]$, the volume of the lattice is $vol\left(L\right)=\sqrt{det(B^{T}B)}$ and the fundamental domain is $P_{1/2}\left(B\right)=\left\{\sum_{i=1}^{n}\alpha_{i}b_{i}|\alpha_{i}\in \left[-\frac{1}{2},\frac{1}{2}\right]\right\}$. The distance between L and vector $v\in R^{m}$ is dist(v,L)=min{∥v−y∥|y∈L}. The *i*-th successive minima $\lambda_{i}\left(L\right)$ is the minimum radius of the ball centered at the origin, which contains *i* linearly independent vectors in the lattice. Let L be an *n*-dimensional lattice; then, the Gaussian heuristic states that $\lambda_{1}\left(L\right)\approx \sqrt{\frac{n}{2\pi e}}vol{\left(L\right)}^{1/n}$.

**Definition 1.** 
*(Shortest vector problem, SVP) For a lattice L, the SVP problem asks to find a nonzero lattice vector v that minimizes the Euclidean nonzero norm ∥v∥.*


**Definition 2.** 
*(Closest vector problem, CVP) For a lattice L, given a target vector $t\in R^{m}$ that is not in L, the CVP problem asks to find a lattice vector v that minimizes the Euclidean norm ∥v−t∥.*


**Definition 3.** 
*(Unique shortest vector problem, Unique-SVP) For a lattice L satisfying $\lambda_{2}\left(L\right)>\gamma \lambda_{1}\left(L\right)$, where γ≫1, the uSVP problem asks to find the shortest nonzero lattice vector.*


**Definition 4.** 
*(Bounded distance decoding, BDD) For a target vector $t\in R^{m}$ that is not in the given lattice L, which satisfies $dist\left(t,L\right)<\gamma \lambda_{1}\left(L\right)$, where γ<1/2, the BDD problem asks to find a nonzero lattice vector v that minimizes the Euclidean norm ∥v−t∥.*


Algorithms for hard problems on lattices usually perform lattice basis reduction as a preprocessing module, because a sufficiently good basis improves the algorithms’ success probability. The LLL (Lenstra–Lenstra–Lovász) algorithm [22] and the BKZ (block–Korkin–Zolotarev) algorithm [23] are two famous basis reduction algorithms.

Before introducing the LLL reduction algorithm, we first explain the Gram–Schmidt orthogonalization. With a lattice basis $B=[b_{1},b_{2},\ldots ,b_{n}]$, one can calculate its Gram–Schmidt orthogonalization $B^{*}=\left[b_{1}^{*},b_{2}^{*},\ldots ,b_{n}^{*}\right]$ by the recursion $b_{1}^{*}=b_{1}$,$b_{i}^{*}=b_{i}-\sum_{j=1}^{i-1}\mu_{i,j}b_{j}^{*}$ for i=2,3,…,n, where the Gram–Schmidt coefficients $\mu_{i,j}=\langle b_{i},b_{j}^{*}\rangle /\langle b_{j}^{*},b_{j}^{*}\rangle$. The LLL algorithm was proposed in 1982, and the formal description of LLL reduction is detailed as shown in Algorithm 1.
**Algorithm 1** LLL algorithm.**Input:** lattice basis $B=\left[b_{1},b_{2},\ldots ,b_{n}\right]\in R^{m\times n}$, a reduction parameter δ.**Output:** a δ-LLL reduced basis 1:Calculate the Gram–Schmidt orthogonalization $B^{*}=\left[b_{1}^{*},b_{2}^{*},\ldots ,b_{n}^{*}\right]$. 2:**for** i = 2, 3,…, n **do** 3:   **for** j = i − 1, i − 2,…, 1 **do** 4:     $b_{i}=b_{i}-c_{i,j}b_{j}$, where $c_{i,j}=\left\lceil \langle b_{i},b_{j}^{*}\rangle /\langle b_{j}^{*},b_{j}^{*}\rangle \right\rfloor$; 5:   **end for** 6:**end for** 7:**if** ∃i, s.t.$\delta ∥b_{i-1}^{*}{∥}^{2}>{∥\mu_{i,i-1}b_{i-1}^{*}+b_{i}^{*}∥}^{2}$ **then** 8:   Swap $b_{i-1}$ and $b_{i}$; 9:   Go to Step 1. 10:**end if** 11:**return** B.

The BKZ algorithm is derived from the KZ (Korkine–Zolotarev) reduction. BKZ uses the block reduction to improve the LLL algorithm and outputs an (δ,β)-BKZ reduced basis. To be specific, the BKZ algorithm runs the enumeration algorithm on the sub-lattice with block size β and obtains its shortest vector. After inserting the shortest vector into the original basis, LLL reduction with parameter δ is applied on the entire basis to remove the linear dependency. BKZ performs the above steps iteratively until the basis is no longer updated.

### 2.2. The LWE Problem

**Definition 5.** 
*(Learning with errors distribution) Let n,q>0 be integers, and α∈{0,1}. Let $s\in Z_{q}^{n}$ be a secret vector. The LWE distribution $\chi_{s,\alpha }$ refers to $\left(a,\langle a,s\rangle +e\right)\in Z_{q}^{n}\times Z_{q}$, where $a\in Z_{q}^{n}$ is uniformly selected randomly and e is a discrete Gaussian error with standard deviation αq.*


**Definition 6.** 
*(Learning with errors problem) Let n,m,q>0 be integers, α>0. Given m samples $(a_{i},\langle a_{i},s\rangle +e_{i}),i=1,2,\ldots ,m$, the search-LWE problem asks to recover the secret vector $s\in Z_{q}^{n}$, and the decision-LWE problem asks to determine whether the samples are sampled according to $\chi_{s,\alpha }$ or the uniform distribution.*


Now, we review some lattice-based methods for analyzing the LWE problem. In general, the decision-LWE can be solved by the short integer solution strategy, and the search-LWE can be attacked by the BDD strategy or the inhomogeneous short integer solution strategy. Now, we mainly describe the decoding method and the primal method in the BDD strategy.

The LWE problem can be written in a matrix form c=As+emodq. Given $q\in Z,c\in Z_{q}^{m},A={\left[a_{1},\ldots ,a_{m}\right]}^{T}\in Z_{q}^{m\times n}$, the problem recovers s. The basic idea of the decoding method is to regard c as the target vector and then use the Nearest Plane algorithm to find the closest vector in $\Lambda_{q}\left(A\right)$. Assuming the basis of $\Lambda_{q}\left(A\right)$ is B, before applying the Babai’s Nearest Plane algorithm, B should be preprocessed to a Gram–Schmidt basis $B^{*}$. The strategy outputs s if and only if e lies in $s+P_{1/2}\left(B^{*}\right)$, which is determined by the quality of the basis. Lindner and Peikert improved Babai’s algorithm by admitting a time/success trade-off. To be specific, in each iteration, the Lindner–Peikert Nearest Plane algorithm chooses several close hyperplanes instead of only the closest hyperplane. The idea stretches $P_{1/2}\left(B^{*}\right)$ to a cube-like shape and amplifies the success probability.

The primal method is to solve LWE by reducing BDD to the Unique-SVP problem using an embedding technique. The embedding method is to construct a (m+1)-dimensional lattice $B^{\prime }=\left[\begin{array}{cc} B & c\\ 0 & t \end{array}\right]$. Obviously, the short vector $\left[-e,t\right]\in Z_{q}^{m+1}$ is in $B^{\prime }$. Therefore, solving the Unique-SVP instance recovers the error vector and the secret vector in passing.

### 2.3. Variational Quantum Algorithm

VQA is a quantum–classical hybrid algorithm that is considered to be implemented on NISQ devices. Therefore, VQA is expected to demonstrate quantum advantages over classical computers when solving some specific problems. The workflow of VQA is shown in Algorithm 2.
**Algorithm 2** VQA algorithm.**Input:** An optimization problem.**Output:** Parameters in the parameterized quantum circuit. 1:Construct the objective function. 2:Construct the parameterized quantum circuit. 3:Prepare the quantum state and measure the expectation value. 4:Use a classical optimizer to determine new parameters. 5:Iterate the procedure in step 3 and 4 until the convergence of the value. 6:**return** the final parameters.

There are four important modules in VQA [24,25]: the objective function refers to the cost function that needs to be minimized; the parameterized quantum circuit refers to a set of unitary operators that manipulate parameters in the optimization process; the measurement scheme calculates the expectation value; the classical optimizer outputs the parameters that minimize the objective function.

First, VQA encodes the problem into an objective function *O*. Let the probability of measuring qubit *q* in state |0〉 be $p^{q}$; then, the objective function of VQA can be expressed as $min_{\theta }O\left(\theta ,\left\{p\left(\theta \right)\right\}\right)$.

Because it is inconvenient to obtain the function value directly by the measurement probability, the expectation value of a Hamiltonian is introduced, and constructing the objective function is equivalent to constructing its corresponding Hamiltonian. The Hamiltonian is a quantum operator that encodes the information of a physical system. Its expectation value corresponds to the energy of a quantum state. The ground state of the Hamiltonian is often used as the minimization target of a VQA problem. In practice, the expectation value of Hamiltonian *H*
$${\langle H\rangle }_{U(\theta )}=\langle 0|U^{†}\left(\theta \right)HU\left(\theta \right)|0\rangle$$
is used to describe the measurement results of the quantum state produced by U(θ). Therefore, the objective function is
$$min_{\theta }O\left(\theta ,{\langle H\rangle }_{U(\theta )}\right).$$

If the objective function is defined more compactly, it can be described as $min_{\theta }{\langle H\rangle }_{U(\theta )}$. The objective functions or cost functions constructed in this paper are all in the compact form.

Second, parameterized quantum circuits are a set of unitary operations that depend on parameters. The parameterized quantum circuit acting on quantum state $|\psi_{0}\rangle$ can be expressed as
$$\left|\psi \left(\theta \right)\rangle =U\left(\theta \right)\right|\psi_{0}\rangle ,$$
where θ are variational parameters.

Most ansatz *U* can be classified as problem-inspired or hardware-efficient. The construction of problem-inspired ansatz requires the information of specific problems. For example, the united coupled cluster ansatz in quantum chemistry is constructed by a parameterized cluster operator T(θ) and acts on the ground state $|\psi_{HF}\rangle$ in the way of $|\psi \left(\theta \right)\rangle =e^{T\left(\theta \right)-T^{{}^{†}}\left(\theta \right)}|\psi_{HF}\rangle$. Ansatz in the QAOA algorithm is also problem-inspired, and its construction is shown in Section 3. Hardware-efficient ansatz is usually expressed as $\prod_{k=1}^{D}U_{k}\left(\theta_{k}\right)W_{k}$, where $\theta =(\theta_{1},\ldots ,\theta_{D})$, $U_{k}\left(\theta_{k}\right)=e^{-i\theta_{k}V_{k}}$ is a unitary operator derived from Hamiltonian $V_{k}$, and $W_{k}$ is an unparametrized unitary operator.

Third, in order to obtain the information of quantum state, we need to measure it in the computational basis and calculate the expectation value of the objective function. The expectation value of the operator $\sigma^{z}$ can be obtained by $\langle \sigma^{z}\rangle =\langle \psi |\sigma^{z}|\psi \rangle ={\left|\alpha \right|}^{2}-{\left|\beta \right|}^{2}$, where ${\left|\alpha \right|}^{2}$ and ${\left|\beta \right|}^{2}$ are the probabilities to measure |ψ〉 in state |0〉 and |1〉. The measurement defined by $\sigma^{x}$ and $\sigma^{y}$ is first transformed into the basis of $\sigma^{z}$ by $\sigma^{x}=R_{y}^{†}\left(\pi /2\right)\sigma^{z}R_{y}\left(\pi /2\right)$, $\sigma^{y}=R_{x}^{†}\left(\pi /2\right)\sigma^{z}R_{x}\left(\pi /2\right)$ and then measured on a $\sigma^{z}$ basis. Any Pauli string is measured in the same way, except that it is measured on each qubit separately.

QAOA and VQE are two quantum variational algorithms, so they can be used to solve optimization problems. Since a quantum circuit is equivalent to a tensor product, it can be represented on a classical computer, and the expectation value of the cost function can be calculated, but the memory it consumes grows exponentially with the size of the problem. For a quantum computer, repeating the preparation of ansatz state and the quantum measurements, the expectation can be obtained. The quantum resources it consumes increase polynomially with the scale of the problem, thus showing its superiority over classical algorithms.

## 3. The Decoding Method for Solving LWE

This section applies the decoding method to solve LWE. When solving BDD, we use QAOA to improve Babai’s Nearest Plane algorithm.

First, construct a q-ary lattice $\Lambda_{q}\left(A\right)=\left\{v\in Z_{q}^{m}|\exists x\in Z^{n},s.t.v\equiv Axmodq\right\}$, whose lattice basis is equivalent to $B={\left[A|qI_{m}\right]}^{T}\in Z^{(m+n)\times m}$. Second, perform elementary row transformations on B and obtain a basis matrix ${\left[b_{1}^{\prime },\ldots ,b_{m}^{\prime }\right]}^{T}\in Z^{m\times m}$. Third, solve CVP with the target vector c, and finally output the closest vector w. The last step is to use the Gaussian elimination to recover $s=A^{-1}w$.

Now, introduce the application of QAOA when improving Babai’s Nearest Plane algorithm. Babai’s Nearest Plane algorithm consists of two steps: first, perform the LLL reduction on the input lattice basis, and then find the linear combination in the reduced basis so that it forms the closest lattice vector to the given target vector. The formal description is detailed as Algorithm 3.

In the loop, $u_{j}=\left\lceil \langle b,b_{j}^{*}\rangle /\langle b_{j}^{*},b_{j}^{*}\rangle \right\rfloor$ only takes one value by the “round to the nearest integer” function. Through experiments, it is found that when the value range is expanded to $\left\{u_{j}+x|x=0,1,-1\right\}$, a better solution is often obtained. In a classical algorithm, the process requires an exponential increase in computation with respect to the lattice dimension *n*. In quantum computing, due to quantum properties, the computing complexity can be greatly reduced. Therefore, we now introduce the method of encoding the random floating in $u_{j}$ in two qubits and solving the optimization problem by QAOA.
**Algorithm 3** Babai’s Nearest Plane algorithm.**Input:** lattice basis $B^{\prime }=\left[b_{1}^{\prime },b_{2}^{\prime },\ldots ,b_{m}^{\prime }\right]\in R^{m\times m}$, target vector $t\in Z^{m}$**Output:** vector $x\in L(B^{\prime })$, which satisfies $∥x-t∥\leq 2^{m/2}dist\left(t,L\left(B^{\prime }\right)\right)$ 1:Perform the LLL reduction on $B^{\prime }$ with parameter δ=3/4. 2:Use the Gram–Schmidt orthogonalization on the reduced basis and obtain $B^{*}=\left[b_{1}^{*},b_{2}^{*},\ldots ,b_{m}^{*}\right]$. 3:b=t. 4:**for** j = m, m − 1,…, 1 **do** 5:   $b=b-u_{j}b_{j}^{\prime }$, where $u_{j}=\left\lceil \langle b,b_{j}^{*}\rangle /\langle b_{j}^{*},b_{j}^{*}\rangle \right\rfloor$; 6:**end for** 7:**return **t−b

First, apply Babai’s Nearest Plane algorithm to calculate the classical optimal solution, that is, the shortest distance vector $b_{op}=\left(b_{op}^{1},b_{op}^{2},\ldots ,b_{op}^{m}\right)$. Then, the result is improved by QAOA. Let the LLL-reduced basis in Babai’s algorithm be $D=[d_{1},d_{2},\ldots ,d_{m}]$, and construct the optimization function
$$F\left(x_{1},x_{2},\ldots ,x_{m}\right)={∥\sum_{i=1}^{m}x_{i}d_{i}-b_{op}∥}^{2},$$
where $x_{i}\in \left\{-1,0,1\right\},i=1,2,\ldots ,m$. It is easy to verify that $F(x_{1},x_{2},\ldots ,x_{m})$ is a non-negative function. Let ${\hat{x}}_{i}=\frac{\sigma_{2i-1}^{z}+\sigma_{2i}^{z}}{2}$, which is a quantum operator encoded in the Pauli-Z basis. The eigenvalues of operator ${\hat{x}}_{i}$ are −1, 0, 1, which exactly encodes the value of the variable $x_{i}$. Therefore, the corresponding problem Hamiltonian is
$$H_{C}=\sum_{j=1}^{m}{\left|\sum_{i=1}^{m}d_{i,j}{\hat{x}}_{i}-b_{op}^{j}I\right|}^{2}.$$
Obviously, for an *m*-dimensional lattice, the number of qubits required to optimize Babai’s algorithm is 2m.

To solve the problem, it is necessary to introduce a mixing Hamiltonian $H_{M}=\sum_{i=1}^{2m}\sigma_{i}^{x}$, where $\sigma_{i}^{x}$ is the Pauli-X operator acting on the ith bit. The quantum circuit of QAOA is defined by the problem Hamiltonian $H_{C}$, the mixing Hamiltonian $H_{M}$ and parameters (γ,β). For *D*-layer QAOA circuits, there are usually 2D variational parameters. The process of using QAOA to solve the optimization problem is shown in Figure 1, and the algorithm description is shown in Algorithm 4.
**Algorithm 4** QAOA solving optimization.**Input:** the problem Hamiltonian $H_{C}$, the mixing Hamiltonian $H_{M}$.**Output:** the ground state $|\Psi_{C}\rangle$ of $H_{C}$. 1:Prepare the quantum register into $|\Psi_{0}{\rangle =|+\rangle }^{⊗m}$. 2:Choose the initial parameters γ, β. Perform $H_{C}$ and $H_{M}$ alternately and obtain |Ψ(γ,β)〉. 3:Measure the quantum registers and calculate the cost function. 4:Repeat Step 2 and Step 3 several times and calculate the expectation value of the cost function. 5:Pass the expectation value and parameters (γ,β) to a classical optimizer. Update the parameters (γ,β). 6:Repeat Steps 2–5 until the result meets a fixed threshold and the parameters are updated to $\left(\gamma^{*},\beta^{*}\right)$. 7:**return** $|\Psi_{C}\rangle =|\Psi \left(\gamma^{*},\beta^{*}\right)\rangle$

Now, we explain the steps in Algorithm 4. Step 1 performs $H^{⊗m}$ on ${|0\rangle }^{⊗m}$, and we obtain ${|+\rangle }^{⊗m}$, which is an eigenvector of the Pauli-X operator.

Step 2 applies operators $e^{-i\gamma_{k}H_{C}}$ and $e^{-i\beta_{k}H_{M}}$, k=1,2,…,D, alternately. So, we generate a variational wave function
(1)$$|\phi \left(\gamma ,\beta \right)\rangle =e^{-i\gamma_{D}H_{C}}e^{-i\beta_{D}H_{M}}\ldots e^{-i\gamma_{1}H_{C}}e^{-i\beta_{1}H_{M}}{|+\rangle }^{⊗m}.$$
The wave function has 2D parameters $\left\{\gamma_{1},\ldots ,\gamma_{D},\beta_{1},\ldots ,\beta_{D}\right\}$.

The expectation value means
(2)$$\langle \Psi \left(\gamma ,\beta \right)|H_{C}|\Psi (\gamma ,\beta \rangle ,$$
which can be obtained by repeatedly preparing |Ψ(γ,β)〉 on the quantum processor and measuring it on a computational basis. Then, the classical computer performs classical optimization algorithms to find the optimal parameter. For example, the optimizers use the gradient descent algorithm to minimize the cost function in an iterative manner. The method calculates the first-order derivative of the function to compute the gradient. Then, it moves in the negative direction of the gradient. The termination condition of the gradient descent method is that the slope of the gradient is below a very small threshold. In the actual experiment, the algorithm is terminated by setting the empirical number of iterations.

In fact, classical optimization problems are often mapped to a simple Hamiltonian, which is diagonal in the computational basis. However, it does not mean that the problem is easy to solve or does not require a quantum solver. First, for example, MaxCut is a classical NP-hard problem, and the design of MaxCut problem Hamiltonian is $H=\sum_{ij}\frac{1}{2}\left(I-\sigma_{i}^{z}\sigma_{j}^{z}\right)$ [16]. In computational complexity theory, P is a set of relatively easy problems, and NP indicates hard problems. If MaxCut can be solved by classical computers easily, then P = NP, which completely overturns the theoretical basis of a range of fields. Second, processing classical optimization by QAOA usually requires a mixing Hamiltonian consisting of $\sigma^{x}$ or $\sigma^{y}$, so quantum computers still work when solving classical optimization problems.

## 4. The Primal Method for Solving LWE

In this section, we propose a quantum primal method for solving LWE, where the Unique-SVP problem is solved by VQE. Although the quantum advantage of solving classical optimization by VQE is not as obvious as it is in quantum chemistry, understanding the evolution of the algorithm process is still crucial for improving algorithms running on classical hardware. We detail the number of qubits required and estimate the quantum resources when attacking the KYBER cryptosystem. With the development of quantum computers, resource estimation can also be used as a direction for comparison with pure classical algorithms.

### 4.1. LWE Algorithm

Algotithm 5 shows the procedure of the LWE algorithm.
**Algorithm 5** The LWE algorithm.**Input:** LWE samples $\left(A,c=As+e\right)\in Z_{q}^{m\times n}\times Z_{q}^{m}$**Output:** secret vector $s\in Z_{q}^{n}$ 1:Construct a q-ary lattice $\Lambda_{q}\left(A\right)=\left\{v\in Z_{q}^{m}|\exists x\in Z^{n},s.t.v\equiv Axmodq\right\}$, whose lattice basis is equivalent to $B={\left[A|qI_{m}\right]}^{T}\in Z^{(m+n)\times m}$. 2:Perform elementary row transformations on B and obtain the lattice basis $B_{1}=\left[\begin{array}{cc} I_{n} & A_{n\times (m-n)}^{{}^{\prime }}\\ 0 & qI_{m-n} \end{array}\right]\in Z^{m\times m}$. 3:Using Kannan’s embedding technique, reduce BDD to Unique-SVP and obtain $B_{2}=\left[\begin{array}{cc} B_{1} & 0\\ c & M \end{array}\right]\in Z^{\left(m+1)\times (m+1\right)}$. 4:Process $B_{2}$ with VQE and derive a short vector e. 5:**return**$s=A^{-1}\left(c-e\right)$

Step 3 expands the q-ary basis by one dimension and embeds the target vector c and the embedding factor *M* into matrix $B_{2}$. When M=∥e∥, there exists $\left(e,-M\right)\in L\left(B_{2}\right)$ [26]. In this case, proposing the first *m* bits of the vector recovers e. In the experiment, we generally take M=1.

Unique-SVP can be seen as a special case of SVP, and step 4 in Algorithm 5 solves SVP by VQE. The detailed description is shown in Algorithm 6.
**Algorithm 6** VQE solving SVP.**Input:** the lattice basis $B={\left[b_{1},\ldots ,b_{m}\right]}^{T}\in Z^{\left(m+1)\times (m+1\right)}$.**Output:** short vector x. 1:Perform BKZ-reduction on B. 2:The SVP problem is encoded to the ground state of the Hamiltonian operator *H*. 3:Construct parameterized quantum circuits. 4:Repeat preparing an ansatz state |Ψ(θ)〉 from the parameterized quantum circuit and measuring it in Pauli-Z basis. Calculate the expectation value C(θ). 5:Pass C(θ) and parameters to a classical optimizer. Update the parameter θ and go to step 4 until the expectation value converges.

The VQE procedure is visualized in Figure 2. Now, we explain the steps in Algorithm 6 in detail. In step 1, the larger the lattice size, the more quantum resources it occupies. In order to reduce the required qubits, a new basis matrix is first obtained by performing the BKZ reduction.

Step 2 constructs the problem Hamiltonian. For Lattice B, SVP is to find a nonzero vector x satisfying $min_{x\in L(B)}∥x∥$. Let the row vector of coefficients be z and z≠0; then, we have x=zB. Let $G=BB^{T}$; then, we have ${∥x∥}^{2}=zBB^{T}z^{T}=zGz^{T}$. According to Algorithm 5, the dimension of the lattice is $m^{\prime }=m+1$. So, the SVP problem is equivalent to
(3)$$min_{x\in L(B)}{∥x∥}^{2}=min_{z\in Z^{m^{\prime }}}\left(\sum_{i=1}^{m^{\prime }}z_{i}^{2}G_{ii}+2\sum_{0\leq i<j\leq m^{\prime }}z_{i}z_{j}G_{ij}\right).$$

Before mapping the SVP problem into a Hamiltonian, we first introduce the method of reducing numbers in the integer interval [−d,d] to a Boolean variable polynomial. Let t=⌊logd⌋, introducing t+1 Boolean variables $\beta_{0},\beta_{1},\beta_{2},\ldots ,\beta_{t}$; the number in the interval can be expressed as $\sum_{i=0}^{t-1}2^{i}\beta_{i}+\left(2d+1-2^{t}\right)\beta_{t}-d$. Therefore, for the coefficient vector z, if each entry satisfies $|z_{i}|\leq d_{i}$, $i=1,2,\ldots ,m^{\prime }$, it can be expressed by Boolean variables $\beta_{i0},\ldots ,\beta_{it_{i}}$. Substituting the Boolean variable polynomials into (3), we have
$$min_{\beta_{10},\ldots ,\beta_{1t_{1}},\ldots ,\beta_{m^{\prime }0},\ldots ,\beta_{m^{\prime }t_{m^{\prime }}}}\left(h+\sum_{ij}h_{ij}\beta_{ij}^{2}+\sum_{ij\neq kl}l_{ij,kl}\beta_{ij}\beta_{kl}\right),$$
where $h,h_{ij},l_{ij,kl}$ are calculated constants. Because $\beta_{ij}$ are Boolean variables, the above equation is equivalent to
(4)$$min_{\beta_{10},\ldots ,\beta_{1t_{1}},\ldots ,\beta_{m^{\prime }0},\ldots ,\beta_{m^{\prime }t_{m^{\prime }}}}\left(h+\sum_{ij}h_{ij}\beta_{ij}+\sum_{ij\neq kl}l_{ij,kl}\beta_{ij}\beta_{kl}\right).$$

In the above formula, it is required to find the parameter vector
(5)$$\beta =(\beta_{10},\ldots ,\beta_{1t_{1}},\ldots ,\beta_{m^{\prime }0},\ldots ,\beta_{m^{\prime }t_{m^{\prime }}})$$
to minimize the function
$$\sum_{ij}h_{ij}\beta_{ij}+\sum_{ij\neq kl}l_{ij,kl}\beta_{ij}\beta_{kl}.$$

Encoding the cost function into a Hamiltonian requires a mapping $\beta_{ij}\to \left(1-\gamma_{ij}\right)/2$, where $\gamma_{ij}\in \left\{-1,1\right\}$. Then, substitute $\gamma_{ij}\to \sigma_{ij}^{z}$ and $1\to I_{ij}$ to obtain the problem Hamiltonian
$$H=\sum_{ij}h_{ij}\frac{I_{ij}-\sigma_{ij}^{z}}{2}+\sum_{ij\neq kl}l_{ij,kl}\frac{I_{i,j}-\sigma_{ij}^{z}}{2}⊗\frac{I_{kl}-\sigma_{kl}^{z}}{2},$$
where $ij,kl\in \{10,\ldots ,1t_{1},m^{\prime }0,\ldots ,m^{\prime }t_{m^{\prime }}\}$ and $\sigma_{i}^{z}$ is the Pauli-Z operator acting on the *i*th bit. The Hamiltonian acts on a Hilbert space spanned by QNum qubits, and it can also be written as a sum over many local interactions.

To find the ground state of *H*, step 3 generates a hardware-efficient trial wavefunction, which is more suitable for available quantum devices [27]. Let $|\Psi \left(\theta \right)\rangle ={\left(U\left(\theta \right)U_{ENT}\right)}^{D}|\Psi_{0}\rangle$ and the reference state is set to |00..0〉. U(θ) are a group of single-qubit rotations determined by rotation angles θ. $U_{ENT}$ are entangling drift operations generating sufficient entanglement. *D* defines the level of the quantum circuit. Obviously, with the increase of *D*, the convergence speed increases, but the fidelity decreases.

Step 4 calculates C(θ)=〈Ψ(θ)|H|Ψ(θ)〉. Each iteration requires measuring *N* times and the cost obtained for the *i*-th time is $C_{i}$. Then, the expectation value is
(6)$$C\left(\theta \right)=\langle \Psi \left(\theta \right)|H|\Psi \left(\theta \right)\rangle =\frac{1}{N}\sum_{i=1}^{N}C_{i}.$$

If the Hilbert space is too large, because the interaction is local, the Hamiltonian can be split into a summation over many terms. The expectation calculations for one term are relatively simple, and we can speed up the computation by parallelizing the quantum expectation-value estimation algorithm [28]. After calculating the expectation of each item on the quantum processor, multiply it by the weight and sum on the classical processor to obtain the final expectation value.

However, the shortest vector is 0 in this algorithm, so the restriction x≠0 needs to be added. The idea is to increase *C* when appearing as 0. We assume that among the *N* measurements, there are $N_{0}$ results that are not 0, $C=\frac{1}{N_{0}}\sum_{i=1}^{N}C_{i}$. Obviously, the larger the $N_{0}$, the smaller the *C*.

Step 5 uses the classical optimization algorithm to update θ until the expectation value converges and the process is similar to QAOA.

We give a toy example to illustrate the process on the quantum processor. For a more convenient description, the LWE dimension is further limited, and the example also supports simple experiments on the IBM quantum system. Let q=3.n=1,m=2. The samples are $s+e_{1}=1mod3,2s+e_{2}=2mod3$. The LLL-reduced matrix after Kannan’s embedding is
$$\left[\begin{array}{ccc} 0 & 0 & 1\\ -1 & 1 & 0\\ 1 & 2 & 0 \end{array}\right].$$

To simplify the model, suppose $z_{i},i=1,2,3$, are already Boolean variables. Then, the SVP problem can be reduced into finding the minimum value of $C=z_{1}+2z_{2}+5z_{3}+2z_{2}z_{3}$. The problem Hamilton is
(7)$$H=4.5I_{1}⊗I_{2}⊗I_{3}-0.5Z_{1}⊗I_{2}⊗I_{3}-1.5I_{1}⊗Z_{2}⊗I_{3}-3I_{1}⊗I_{2}⊗Z_{3}+0.5I_{1}⊗Z_{2}⊗Z_{3}.$$

Now, construct a hardware-efficient Ansatz consisting of several parameterized single-qubit rotation operations and controlled-NOT gates. Using the parameterized circuit shown in Figure 3, any 3-qubit quantum state |Ψ(θ)〉 can be prepared, and different quantum states can be output by adjusting the six parameters.

After preparing the ansatz state and measuring it repeatedly, we calculate the expectation value. Then, we perform the optimization process on the classical processor. Iterate the above process, and finally, the parameters corresponding to the optimal result are (0,π,0,0,0,π) and $\left[z_{1},z_{2},z_{3}\right]=\left[1,0,0\right]$. So, $\left[e_{1},e_{2}\right]=\left[0,0\right]$, s=1.

### 4.2. Algorithm Analysis

First, we analyze the range of $d_{i}$ in the restriction condition $|z_{i}|<d_{i},i=1,2,\ldots ,m^{\prime }$. Let $\tilde{B}={\left(B^{-1}\right)}^{T}={\left[{\tilde{b}}_{1},\ldots ,{\tilde{b}}_{m^{\prime }}\right]}^{T}$; then, there exists $\langle b_{i},{\tilde{b}}_{i}\rangle =\{\begin{array}{ccc} 1 & & i=j\\ 0 & & i\neq j \end{array}$. Let the shortest vector $v=\sum_{i=1}^{m^{\prime }}t_{i}b_{i}$; then, $|\langle v,{\tilde{b}}_{i}\rangle \left|=\right|t_{i}|\leq ∥v∥∥{\tilde{b}}_{i}∥$. Due to the Gaussian heuristic, $∥v∥=\sqrt{\frac{m^{\prime }}{2\pi e}}vol{\left(L\right)}^{1/m^{\prime }}$, we have $|t_{i}|\leq \sqrt{\frac{m^{\prime }}{2\pi e}}vol{\left(L\right)}^{1/m^{\prime }}∥{\tilde{b}}_{i}∥$.

For an $m^{\prime }$-dimensional matrix B, its orthogonality defect $\delta \left(B\right)=\frac{\prod_{i=1}^{m^{\prime }}∥b_{i}∥}{\left|det(B)\right|}$. Obviously, for B, there exists δ(B)≥1 and δ(B)=1 if and only if B is an orthogonal matrix. Therefore, the total number of qubits can be expressed as
(8)$$QNum=\sum_{i=1}^{m^{\prime }}\left(\left\lfloor logd_{i}\right\rfloor +1\right)\leq m^{\prime }+log\left(d_{1}d_{2}\ldots d_{m^{\prime }}\right),$$
where $log\left(d_{1}d_{2}\ldots d_{m^{\prime }}\right)\leq 0.5m^{\prime }log\left(\frac{m^{\prime }}{2\pi e}\right)+log(vol\left(L\right)\prod_{i=1}^{m^{\prime }}∥{\tilde{b}}_{i}∥)=0.5m^{\prime }log\left(\frac{m^{\prime }}{2\pi e}\right)+log\left(\delta \left(\tilde{B}\right)\right)$. For a KZ-reduced matrix B, its orthogonality defect satisfies [29]
$$\delta \left(B\right)\leq {\left(\frac{1}{8}m^{\prime }+\frac{6}{5}\right)}^{m^{\prime }/2}\left(\prod_{i=1}^{m^{\prime }}\frac{\sqrt{i+3}}{2}\right)\leq {\left(\frac{1}{8}m^{\prime }+\frac{6}{5}\right)}^{m^{\prime }/2}{\left(m^{\prime }+3\right)}^{m^{\prime }/2}{\left(\frac{1}{2}\right)}^{m^{\prime }}.$$
So,
$$log\left(\delta \left(\tilde{B}\right)\right)\leq \frac{m^{\prime }}{2}log\left(\frac{1}{8}m^{\prime }+\frac{6}{5}\right)+\frac{m^{\prime }}{2}log\left(m^{\prime }+3\right)-m^{\prime }\leq m^{\prime }log\left(m^{\prime }+3\right)-m^{\prime }.$$
Substituting into Equation (Equation 8), we have
(9)$$\begin{array}{c} QNum\leq m^{\prime }+\left(\frac{m^{\prime }}{2}log\left(m^{\prime }\right)-\frac{m^{\prime }}{2}log\left(2\pi e\right)+m^{\prime }log\left(m^{\prime }+3\right)-m^{\prime }\right)\\ =\frac{m^{\prime }}{2}log\left(m^{\prime }\right)-\frac{m^{\prime }}{2}log\left(2\pi e\right)+m^{\prime }log\left(m^{\prime }+3\right) \end{array}$$
Therefore, the maximum number of qubits is $O(m^{\prime }logm^{\prime })$. Now, we review the value of $d_{i},i=1,2,\ldots ,m^{\prime }$ when using VQE for enumeration. In practice, each $z_{i}$ is represented by $QNum/m^{\prime }$ qubits and the range of $d_{i}$ is $\left[2^{(QNum/m^{\prime })-1},2^{\left(QNum/m^{\prime }\right)}-1\right]$, where $d_{i}\in Z$.

In Kannan’s embedding, the lattice dimension is m+1, where *m* is the sample number. In most cases, an LWE-based scheme produces only m=poly(n) LWE samples (and the polynomial bound can be as small as m=Θ(n)). In the LWE-based cryptosystem proposed in paper [12], $m=\sqrt{nlg(q)/lg(\delta )}$ and δ here means the root-Hermite factor. The theoretical worst-case reduction for LWE requires $\alpha q\geq 2\sqrt{n}$ [4], so we set $\alpha q=2\sqrt{n}$. Now, we analyze the average number of qubits required, and its LWE parameters are shown in Table 1.

There are 4 groups of parameters in the table. For each group, 10 experiments are performed, and the average value of the cost function *C* is obtained. Finally, we calculate the average number of qubits required, and the result is illustrated in Figure 4. The four curves with different colors represent that the preprocessing method for the lattice basis is LLL, BKZ-20, BKZ-40 and BKZ-80, respectively. By the regression analysis, taking BKZ-20 as an example, we have
QNum=92.54nlogn−612.27n+1343.8logn−1234.37.
For example, for a 40-dimensional LWE problem, the maximum number of qubits required is 1126, which is a scale that is considered achievable in the near future. With the further development of quantum computers, LWE with larger dimensions can also be solved successively.

### 4.3. Attacks on Existing Cryptosystems

In this section, we calculate the number of qubits required for a VQE attack on the KYBER cryptosystem. KYBER is a key encapsulation mechanism based on the module-LWE problem, which means it is based on Ring $R=Z\left[X\right]/\left(X^{256}+1\right)$. KYBER has three modes to satisfy 128/192/256-bit security, respectively. The parameters are listed in Table 2.

In the table, n,k,q represents the maximum degree of polynomial, the number of polynomials in each vector and the modulus. The most famous attack on the MLWE problem does not utilize the special structure of a lattice, so we still analyze it as an LWE problem. Paper [7] mentioned that the number of samples is between 0 and (k+1)n. To analyze the worst case, let m=(k+1)n. Therefore, in the primal attack, the lattice dimension d=m+1=(k+1)n+1. Using the conclusion in Section 4.2, for the above three parameter settings, the required maximum qubits are 13,768, 19,538, and 25,482, respectively.

Although the quantum computers made at this stage are all NISQ devices, after IBM launched the 127-QubitEagle processor in 2021, it plans to launch the 1121-QubitCondor processor in 2023. At the same time, the IBM team also fully considered the future million-qubit system when designing the world’s largest dilution refrigerator “Goldeneye”, which is an important part of the IBM’s roadmap for scaling quantum technology.

## 5. Algorithm Implementation and Experimental Results

### 5.1. Using QAOA Algorithm to Improve the Decoding Method

In this section, we discuss the quantum advantage of the algorithm introduced in Section 3. Since it is difficult to estimate the computing complexity of QAOA, the QAOA process is regarded as a black box; that is, it is assumed that QAOA returns the solution to the optimization problem in a limited time. Now, without considering the actual complexity of QAOA, we only analyze the results of the algorithm through small-scale experiments.

The LWE instance is $\left(A,c=As+e\right)\in Z_{q}^{m\times n}\times Z_{q}^{m}$. Thus, after reducing to the BDD problem, the target vector is c. Algorithm 3 outputs a classical closest vector w, and the error vector can be obtained by e=c−w. Then, Algorithm 4 updates vectors w and e by QAOA. The result quality r=∥e∥, which means the norm of the error vector. It is obvious that the smaller the *r*, the higher the quality.

Taking the dimension of the secret vector as n=3 and n=5, the experiment generates 50 groups of random LWE samples, respectively. Each group forms an LWE instance. For each instance, after obtaining the closest vector by Babai’s algorithm and calculating the result quality *r*, we use QAOA for optimization to obtain a new approximate closest vector and calculate the quality. Figure 5 shows the comparison of *r* between classical solutions and solutions after quantum optimization when n=3, and Figure 6 illustrates the comparison when n=5.

In Figure 5 and Figure 6, the horizontal axis represents 50 groups of random samples, and the vertical axis represents the result quality *r*. The red columns represent the results of the classical Babai’s algorithm, and the blue columns represent the results after quantum optimization. According to the definition r=∥e∥, a smaller *r* indicates a closer vector and higher quality. As evident in the figures, quantum results have higher quality than classical results in many cases, while in other cases, the results are the same. Therefore, the conclusion that can be drawn from the experiment is that quantum results obtained by QAOA are no worse than their classical counterparts.

### 5.2. Using VQE Algorithm to Realize the Primal Method

In this section, we present the experiments of solving LWE by the primal method. When quantum simulation is performed in a classical computer, the underlying quantum simulation uses QuSET [30], and the front-end interface to implement the algorithm uses C++. In the experiment, better results can be obtained by using the Conditional Vale at Risk (CVaR) method [31]. Specifically, assuming $C_{1},C_{2},\ldots C_{n}$ are sorted in non-decreasing order and in each loop, $C=\frac{1}{\left\lceil pN\right\rceil }\sum_{i=1}^{\left\lceil pN\right\rceil }C_{i}$, where 0<p<1. Paper [21] proposes that p=0.175 gives better results.

On the simulation platform, due to memory constraints, the maximum lattice dimension does not exceed 30, which means the LWE dimension *n* is much smaller than 30. If the input lattice matrix already contains the shortest vector, since the initial parameters of VQE are random and the algorithm still outputs the shortest vector after several iterations, it verifies the correctness of the algorithm. Therefore, when the VQE input is the reduced basis or the shortest vector can be obtained by simple vector addition or subtraction of the input matrix, the solution obtained by VQE is the same as that of the classical algorithm.

When the input is an arbitrary basis, the actual experimental results of the VQE are of poorer quality. The reason is that the simulation platform occupies classical memory, and the qubits for representing entries of the coefficient vector are limited. So, the correct coefficient vector cannot be accurately obtained. As the number of available qubits increases in the future, its coefficient representation will become more and more accurate, and the solution quality of the VQE algorithm will be better.

## 6. Discussion and Conclusions

VQA uses a classical optimizer to train parameterized quantum circuits, and it is one of the most promising quantum algorithms to achieve quantum supremacy. When researchers envision applications for quantum computers, it is almost impossible to bypass VQA algorithms. In this paper, we first present two LWE attacking tools, using QAOA to improve Babai’s algorithm when solving BDD and utilizing VQE to solve Unique-SVP. The two algorithms combine classical optimization techniques and variational quantum techniques, providing ideas for solving LWE when the quantum resources are limited. Second, we estimate the number of qubits required for both algorithms. Third, for the two algorithms, experimental simulations are carried out, respectively. The experimental results show that for the first algorithm, QAOA improves the result quality of classical algorithms, and for the second algorithm, when the memory is large enough, the quality of quantum solutions is at least comparable to that of the classical solutions. How to further reduce the number of qubits by using the structure of the modular lattice is the direction that needs to be studied in the future.

## Figures and Tables

**Figure 1 entropy-24-01428-f001:**
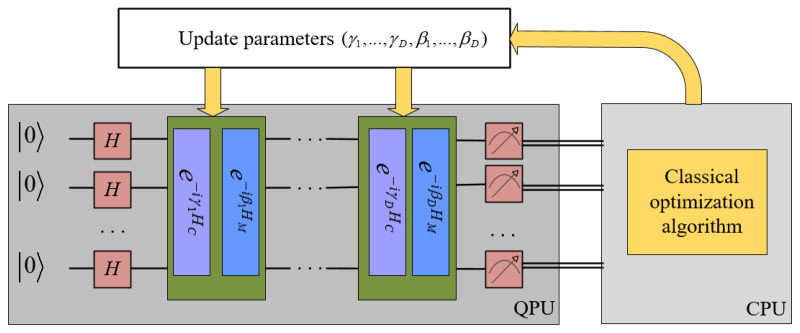
A schematic description of the VQE.

**Figure 2 entropy-24-01428-f002:**
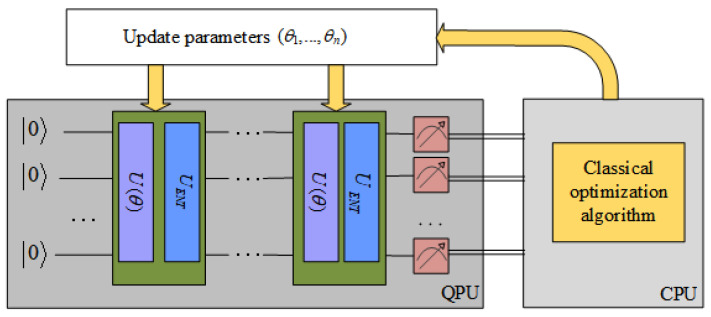
A schematic description of the VQE.

**Figure 3 entropy-24-01428-f003:**
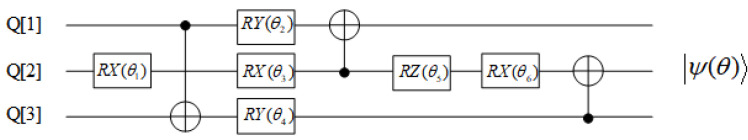
Quantum circuit for 3 qubits.

**Figure 4 entropy-24-01428-f004:**
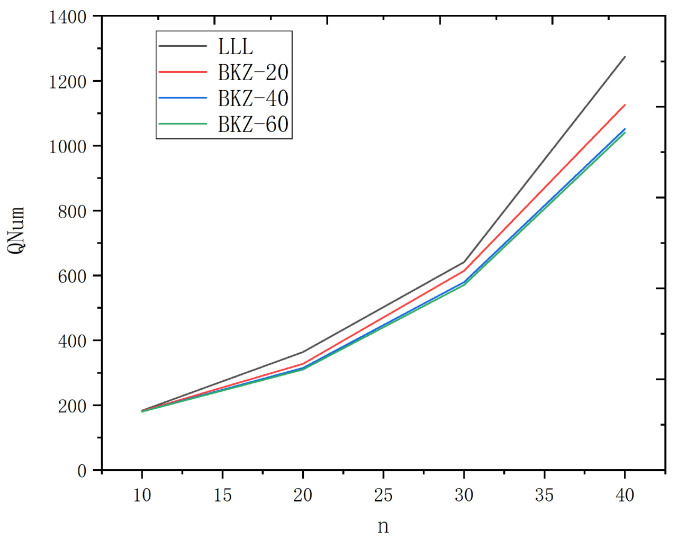
Average number of qubits required for different LWE dimensions.

**Figure 5 entropy-24-01428-f005:**
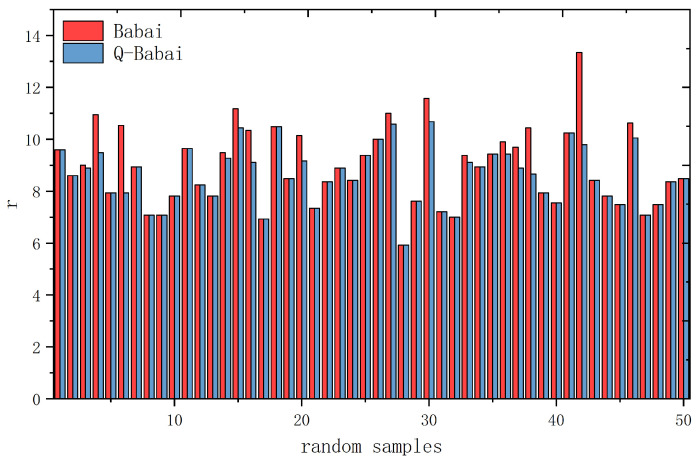
Quantum advantage demonstration of 50 random lattice samples when n=3.

**Figure 6 entropy-24-01428-f006:**
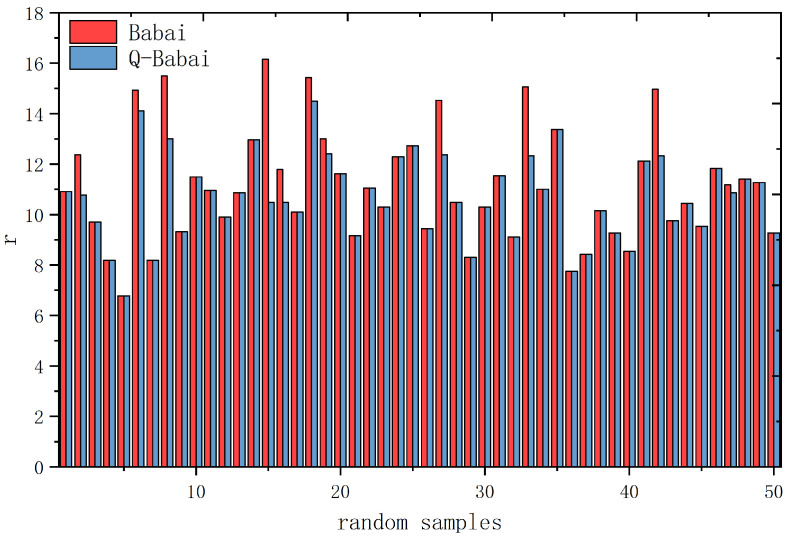
Quantum advantage demonstration of 50 random lattice samples when n=5.

**Table 1 entropy-24-01428-t001:** LWE parameters.

meirent	*n*	10	20	30	40
	*q*	2053	2053	2053	2053
	αq	6.3246	8.9444	10.954	12.649
	*m*	34	65	91	127
	δ	1.069	1.0365	1.0280	1.0191

**Table 2 entropy-24-01428-t002:** KYBER parameters.

	*n*	*k*	*q*
KYBER512	256	2	3329
KYBER768	256	3	3329
KYBER1024	256	4	3329

## Data Availability

The data presented in this study are available within the article.

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
