# Peer review of "Using Variational Quantum Algorithm to Solve the LWE Problem"

_entropy, 2022, doi:10.3390/e24101428_

Round 1
Reviewer 1 Report
The authors consider scenarios in which currently available intermediate scale quantum computers may be advantageous in solving the learning with errors problem. The paper starts with a review of the lattice theory, then the authors show how a quantum adaptation optimization algorithm (QAOA) may be used to improve a solution in the form of the classical closest vector (revealed by the Babai's nearest plane algorithm). In section 5 the authors show how to use QAOA to improve the decoding method. The problems posed and approaches are interesting, however, the paper has some essential drawbacks. They are the following:
It is not clear how the Hamiltonian before line 154 appears. Nor is it clear why operator X has the form as in line 154. The Hilbert space, where H acts, is not clear either. How is the optimization performed for a desired quantity on a quantum computer?
Same as in item 1 for Hamiltonian H at page 6. Additionally, operators Z are not explained.
Do the terms in the Hamiltonian at page 6 commute? If so, then the problem can be solved without a need to use a quantum computer.
Explanation at the end of page 8 is not transparent and must be improved. A formula for the closest vector quality r is needed.
Terms in Hamiltonian at lines 272-273 commute so it is not clear why there is a quantum advantage.
Presentation must be substantially improved too: there are many mistakes in English, punctuation around displayed formulae is to be corrected, a list of abbreviations is highly needed, the number of abbreviations used in abstract is 7 (too many), abbreviation LLL is not explained.
My recommendation is that the paper must be fully revised before it can become suitable for publication in Entropy.
Reviewer 2 Report
Please find the comments and suggestions in the attachment.

Author Response
Not the comments on this article.
Reviewer 3 Report
This paper studies the problem of how to solve the LWE (learning with errors problem) by using VQA, and proposes some ideas of using VQA to solve the LWE problem. However, there are key problems to be addressed before it can be considered for publication.
1. Firstly, it is very important. In Section 2.3, Variational quantum algorithm is recalled, but it is very not clear and I strongly suggest the authors describe the detailed procedure of VQE step-by-step and give corresponding mathematical equations or formulas for each step.
2. It is also very important. The correctness and complexity of the algorithms proposed should be reformulated by means of some theorems together with corresponding proofs mathematically.
3. On page 2, there are some mistakes in the section 2.1, where the dimension of the matrix multiplication does not match.
4. On page 3, in Definition 7, the bracket of the vector is wrong.
5. On Page 4, Section 3 has some mistakes, the dimensions of them does not match.
6. On Page 5, in Algorithm 2, the dimensions of the matrix doesn’t match.
7. On Page 5, in expression 1, concerning the dimensions of expression (1), z belongs to ,and from the last column it seems that we cannot get d.
8. On Page 6, the inequality of QNum lacks “-n”.
On Page 9, the author should write “VQE” instead of “Q
Round 2
Reviewer 1 Report
The authors have carefully and profoundly addressed all the comments and criticisms I raised in the first review. I can recommend the paper for publication in Entropy.
Author Response
thank you very much for your review.
Reviewer 3 Report
The paper has been improved, but the procedure of VQA in Section 2.3 is still not very clear, so I suggest further improvement with more details be done. After that I think the paper can be accepted for publication.
